# Cyclopentadienyl coordination induces unexpected ionic Am−N bonding in an americium bipyridyl complex

Brian N. Long [1], María J. Beltrán-Leiva [1], Cristian Celis-Barros [1], Joseph M. Sperling [1], Todd N. Poe [1], Ryan E. Baumbach [2], Cory J. Windorff [1,3] & Thomas E. Albrecht-Schönzart [1✉]

Variations in bonding between trivalent lanthanides and actinides is critical for reprocessing spent nuclear fuel. The ability to tune bonding and the coordination environment in these trivalent systems is a key factor in identifying a solution for separating lanthanides and actinides. Coordination of $4,4'-$bipyridine $(4,4'-bpy)$ and trimethylsilylcyclopentadienide (Cp') to americium introduces unexpectedly ionic Am−N bonding character and unique spectroscopic properties. Here we report the structural characterization of $(Cp'_3Am)_2(\mu - 4,4'-bpy)$ and its lanthanide analogue, $(Cp'_3Nd)_2(\mu - 4,4'-bpy)$, by single-crystal X-ray diffraction. Spectroscopic techniques in both solid and solution phase are performed in conjunction with theoretical calculations to probe the effects the unique coordination environment has on the electronic structure.

[1] Department of Chemistry and Biochemistry, Florida State University, 95 Chieftan Way, Tallahassee, FL 32306, USA. [2] National High Magnetic Field Laboratory, 1800 E. Paul Dirac Drive, Tallahassee, FL 32310, USA. [3] Department of Chemistry and Biochemistry, New Mexico State University, MSC 3C, PO box 30001 Las Cruces, NM 88003, USA. ✉email: talbrechtschoenzart@gmail.com

The separation of minor trivalent actinides (An³⁺) from their lanthanide (Ln³⁺) counterparts is a crucial step in the reprocessing of spent nuclear fuels; however, this process remains a great challenge owing to similarities in oxidation state and ionic radii[1-3]. Despite these resemblances, prominent differences in the bonding of these elements are observed because of the greater radial extension of the 5f shell in the actinides, compared to the 4f shell of the lanthanides[4-6]. Covalency is an important aspect of bonding in actinides, largely driven by the hard/soft nature of the ligand and the formal oxidation state of the actinide ion[1,5]. Therefore, selective An/Ln separations can be achieved utilizing soft donor ligands, which have demonstrated greater selectivity for actinides over lanthanides due to an increased degree of covalency[7-10].

In particular, Am−N bonding is traditionally expected to exhibit a small degree of covalency owing to π-backbonding between the accessible 5f electrons of americium and π* orbitals of the coordinated, nitrogen-based ligand[1,11,12]. However, manipulation of bond lengths via the introduction of a crowded coordination environment can be applied, directly impacting bonding properties and establishing opportunities to tune the degree of covalency in Am−N systems[13]. Pyridine-based nitrogen-donor ligands, such as 4,4′-bipyridine (4,4′-bpy) fall within this category, and can additionally be used in the synthesis of bridged multinuclear metal complexes[11,14-16]. The ability of 4,4′-bpy to form bimetallic complexes coupled with its known impact on the redox nature of coordinated metals leads to questions about electronic communication between the metal centers, as well as unusual magnetic properties[14,15].

Alternatively, actinide−carbon bonding, particularly in transplutonium elements, is exceedingly under-characterized considering the impact that organoactinide chemistry has built on understanding bonding within the f-block[17,18]. Historically, cyclopentadienide (Cp⁻ = C₅H₅) was used to synthesize the first transuranic organometallic complexes, and its derivatives, Cpᵗᵉᵗ (C₅Me₄H), Cp′ (C₅H₄SiMe₃), and Cp″ [C₅H₃(SiMe₃)₂] have paved the way for solving the unusual bonding and redox chemistry observed in these elements[17,19-36]. Recent advances in transuranic synthesis have yielded the structural characterization of organometallic plutonium and americium complexes[17,19,20]. Single-crystal characterization of an organoamericium complex, [(C₅Me₄H)₃Am], was obtained in 2019, and there is no published organometallic single-crystal characterization for elements heavier than americium to date[17]. However, Cp′ has not yet been applied to americium chemistry due to its formation of highly soluble products in common glovebox solvents, leading to difficult crystallization. To overcome this inconvenience, the bridging ligand 4,4′-bipyridine (4,4′-bpy) can be added to decrease the overall solubility of the complexes and allow for swift, room temperature crystallization suitable for single-crystal X-ray diffraction studies.

In this work, we demonstrate how utilizing Cp′ to tune Am−N and Am−C bonding properties opens a door to the characterization of a family of transuranic organometallic systems and to a deeper comprehension of actinide bonding from both fundamental and applicable viewpoints. Here we report the synthesis and characterization of a multinuclear organometallic americium complex and its neodymium analog (Fig. 1). The synthesis of (Cp′₃Nd)₂(μ-4,4′-bpy) (1-Nd) serves as a synthetic analog to americium, as well as to provide differences in bonding between the 4f and 5f series[37]. The synthesis of (Cp′₃Am)₂(μ-4,4′-bpy) (1-Am) displays distinctively ionic M−N bonding as well as valuable information regarding the intricate nature of covalency in the f-block. Furthermore, the characterization of multinuclear systems in this study lays the foundation for future redox and charge transfer experiments with transplutonium elements.

## Results and discussion

The synthesis of Cp′₃Am was achieved by means of established synthetic methods for the dehydration of actinides using Me₃Si−Br followed by the salt metathesis reaction with KCp′ to form a putative Cp′₃Am residue[17,20,38]. Modeled on the structure of Mehdoui's (Cp′₃U)₂(μ-4,4′-bpy), the addition of 4,4′-bpy to Cp′₃M in toluene results in an immediate precipitate[39]. Heating the slurry to gentle boil yields a solution in which block-shaped crystals were grown upon slow cooling to room temperature and allowing the solution to rest overnight. The compounds were confirmed to be (Cp′₃M)₂(μ-4,4′-bpy) (M = Nd, Am), by single-crystal X-ray crystallography. Both products are isomorphous, crystallizing in the triclinic space group P1̄, and possess an inversion center in the middle of the 4,4′-bpy.

Each metal site is coordinated by three Cp′ rings with a bridging 4,4′-bpy forming a bimetallic structure (Fig. 2). The coordination of 4,4′-bpy causes the Cp′ rings to bend back forming a pseudo-tetrahedral structure. The M−C bond distances range from 2.739(3) to 2.899(3) Å and from 2.761(2) to 2.908(2) Å in 1-Am and 1-Nd, respectively (Supplementary Table 4). Metal carbon bonds of 1-Nd and 1-Am possess average M–C bond lengths of 2.813(2) and 2.795(3) Å. The M−Cent (Cent = Cp′ centroid) distances are 2.506(3), 2.521(3), and 2.544(3) Å for 1-Am, and 2.525(2), 2.542(2), and 2.561(2) Å for 1-Nd. Average metal centroid bond distances of 1-Nd and 1-Am are 2.543(2) and 2.524(3) Å, respectively. These average distances in 1-Nd, 2.813(2) and 2.543(2) Å, are within error of the literature values of 2.806(9) and 2.548(13) Å determined by Deacon et al. in Cp₃Nd(py) (py = pyridine), and similar to the average Nd−Cent bond lengths reported by Minasian et al. in Cp′₃M−ECp* (E = Ga, Cp* = C₅Me₅) of 2.518(3) Å; however, they are shorter than those reported in Cp′₃Ce(py) as anticipated by common lanthanide trends[35,36,40]. The average M−Cent distance observed in 1-Am is shorter than those reported in similar multinuclear organometallic actinide systems, (Cp″₃Th)₂(μ-4,4′-bpy), 2.594(5) Å, and (Cp′₃U)₂(μ-4,4′-bpy), 2.540(10) Å[14,39]. Surprisingly, the average Am–C length is substantially shorter than Th–C despite the tendency of early actinides to reduce the bridging ligand and coordinate in the tetravalent state[14]. The average 1-Nd M−Cent is larger than the reported M−Cent distance for Cp′₃Nd of 2.4885(3) Å[36]. The average metal centroid distance of 1-Am, 2.524 Å, slightly varies with the literature value

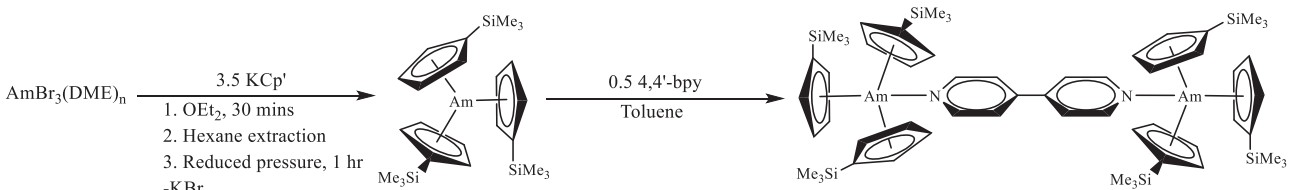

**Fig. 1 Synthesis of the putative Cp′₃Am precursor and (Cp′₃Am)₂(μ-4,4′-bpy), 1−Am.** Addition of 0.5 equivalents of 4,4′-bpy results in a bridged, multinuclear system.

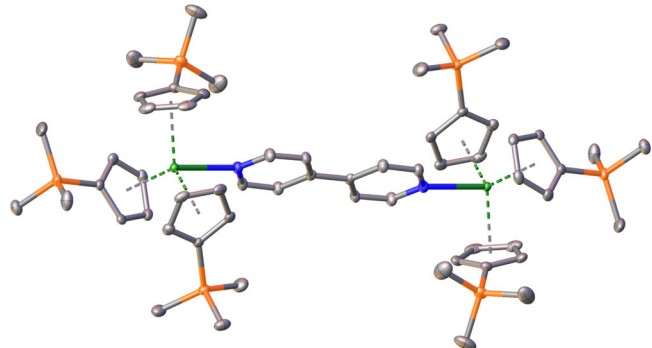

**Fig. 2 Structure of (Cp′₃M)₂(μ-4,4′-bpy) (M = Nd, Am), 1-Am, modeled with thermal ellipsoids at 50% probability. Hydrogen omitted for clarity.** Green = Nd, Am, blue = nitrogen, gray = carbon, and orange = silicon.

of 2.5174 Å for $Cp^{tet}_3Am$, likely due to steric effects of using $Cp′$ over $Cp^{tet}$ and an extra coordination site from 4,4′-bpy[17].

The metal nitrogen distance for **1-Nd** is 2.6482(16) Å versus the 2.618(3) Å of **1-Am**. Overall, **1-Am** exhibits slightly, but statistically significant, shorter M−N bond lengths than that of its lanthanide analog, **1-Nd**. The M−N bond distance in **1-Nd** is similar to that observed in $Cp_3Nd(py)$, 2.668(5) Å, and expectedly shorter than that of $Cp′_3Ce(py)$, 2.704(4) Å[35,40]. **1-Nd** possesses a slightly larger M−N length in comparison $[M(TpyNO_2)(NO_3)_3(H_2O)]·THF$ (M = Nd, Am; $TpyNO_2$ = 4′-nitrophenyl terpyridyl; THF = tetrahydrofuran), with average Nd−N distance, 2.601(2) Å[11]. Similarly, the M−N distance of **1-Am**, 2.618(3) Å, is larger than the reported Am−N distance observed in $[M(TpyNO_2)(NO_3)_3(H_2O)]·THF$, 2.585(2) Å[11]. The same trend is observed in reference to $Am(HDPA)_3$ (HDPA = 2,6-pyridinedicarboxylic acid), which reports an average Am−N distance of 2.564(4) Å and 2.554(4) Å for its Δ and Λ enantiomers, respectively[41]. Assuming that these pyridine-based ligands are valid comparisons, the greater M−N bond distances observed in **1-Nd** and **1-Am** are likely due to steric competition between the Cp′ rings and the bridging 4,4′-bpy. Immediate discrepancies are observed between **1-Am** and early actinide complexes, $(Cp″_3Th)_2(μ-4,4′-bpy)$ and $(Cp′_3U)_2(μ-4,4′-bpy)$, reporting M−N lengths of 2.362(4) and 2.626(7) Å, respectively[14,39]. While Th−N is notably shorter due to the doubly-reduced 4,4′-bipyridine, M−N distances in **1-Am** and $(Cp′_3U)_2(μ-4,4′-bpy)$ are within error, likely owing to the more prevalent ionic interaction observed in the Am−N bond.

The average Cent−M−Cent angles are 117.34° (std. dev. = 0.64°) and 117.35° (std. dev. = 0.79°) for **1-Am** and **1-Nd**, respectively. Compared to the reported values of 119.91 ° determined in Goodwin et al. $Cp^{tet}_3Am$ and 119.81° in Minasian et al. $Cp′_3Nd$, **1-Am** and **1-Nd** possess smaller Cent−M−Cent angles[17,36]. Furthermore, these values consistent with reported angles observed in aforementioned similar systems: 117.373° (std. dev. 2.72°) in $Cp′_3Ce(py)$[40], 117.5° (std. dev. 0.26°) in $Cp_3Nd(py)$[35], and 117.326° (std. dev. 0.92°) in $(Cp′_3U)_2(μ-4,4′-bpy)$[39]. Unexpectedly, the average angle is slightly greater than that reported in $(Cp″_3Th)_2(μ-4,4′-bpy)$, as with the significantly shorter Th−N distance, one would expect a greater shift in ring position owing to a crowded coordination environment[14]. Further coordination comparison can be found in the supplementary information, Supplementary Table 4.

The $^1H$ NMR spectra of **1−Nd** and **1−Am** (Supplementary Figs. 20 and 22) possess resonances expected for trimethylsilyl groups, Cp−H, and 4,4′-bpy. Broadening of these peaks is observed, as is particularly defined in **1−Nd**. Additionally, the 4,4′-bpy resonances experience a shift compared to reported values for pure 4,4′-bpy, indicating coordination of the bridge to

the metal in solution[42]. Magnetic susceptibility measurements via the Evans Method on $Cp′_3Nd$ and **1-Nd** were completed at 293.4 K, yielding a $μ_{eff}$ of 3.75$μ_B$ and 3.47$μ_B$, respectively (Supplementary Figs. 23 and 24), and are comparable to previously reported $Cp″_3Nd$[43,44]. Solid-state variable temperature magnetic susceptibility measurements on crystals of **1-Nd** (Supplementary Fig. 25), from $T = 1.8 – 300$ K, yielded $μ_{eff} = 3.05μ_B$. These results confirm the expected shift in the effective magnetic moments due to the coordination of the 4,4′-bpy bridge.

Solid-state UV–vis–NIR spectra of **1-Nd** and **1-Am** (Fig. 3) were collected at 20 and −180 °C between 350 nm (ca. 28,571 cm$^{-1}$) and 1700 nm (ca. 5,882 cm$^{-1}$). Intense splitting and shifting of the fingerprint f–f transitions to lower energies is observed compared to those of the reported Nd$^{3+}$ and Am$^{3+}$, respectively[45,46]. It is possible that the decrease in electron–electron repulsion between valence electrons upon the coordination of Cp′ and the lowering of symmetry resulting from the coordination of 4,4′-bpy may be the most reasonable explanation for this. Given the fact that intermediate coupling better describes the actinides[47,48], transitions have been assigned according to their total angular momentum, J.

**1-Nd** possesses a charge transfer (CT) band from 400 nm (ca. 25,000 cm$^{-1}$) to 500 nm (ca. 20,000 cm$^{-1}$) shown in Fig. 3. A hypersensitive f–f transition (J = 5/2) is observed at 597 nm (ca. 16,750 cm$^{-1}$), as well as characteristic f–f transitions (J = 5/2, 9/2) at 742 nm (ca. 13,477 cm$^{-1}$) and 801 nm (ca. 12,484 cm$^{-1}$)[49]. **1-Am** has a broad CT band between 350 nm (ca. 28,571 cm$^{-1}$) and 515 nm (ca. 19,417 cm$^{-1}$), slightly overlapping the hypersensitive f–f transition (J = 6) at 527 nm (ca. 18,975 cm$^{-1}$) and 540 nm (ca. 18,519 cm$^{-1}$) shown in Fig. 3[49,50]. Significant splitting of the 785 nm (ca. 12,739 cm$^{-1}$) transition (J = 6) is observed, along with transitions at from 837 to 924 nm (ca. 11,947–10,823 cm$^{-1}$) (J = 5), 924–1121 nm (ca. 10,823–8921 cm$^{-1}$) (J = 4), and 1274–1450 nm (ca. 7849–6897 cm$^{-1}$) (J = 3) typically not observed in Am$^{3+}$ absorption spectroscopy[41,51].

Solution phase UV-vis-NIR spectra of **1-Nd** (Supplementary Fig. 14) were collected at room temperature between 300 nm (ca. 33,333 cm$^{-1}$) and 1000 nm (ca. 10,000 cm$^{-1}$) and spectra of **1-Am** (Supplementary Fig. 17) were collected from 300 nm (ca. 33,333 cm$^{-1}$) to 1700 nm (ca. 5882 cm$^{-1}$). Considerable splitting and shifting of the fingerprint f–f transitions to lower energy is again seen in both complexes. The spectrum of **1-Am** (Supplementary Fig. 17) shows that the characteristic 503 nm hypersensitive transition of Am$^{3+}$ in HClO$_4$ is split and shifted to lower energies, 522 nm (ca. 19,157 cm$^{-1}$) and 541 nm (ca. 18,484 cm$^{-1}$)[45]. The CT band is narrow in comparison to the solid-state spectrum, beginning at roughly 400 nm (ca. 25,000 cm$^{-1}$). Less overlap between the CT band and the 522 nm (ca. 19,157 cm$^{-1}$) transition is observed, revealing transitions from 400 to 500 nm (ca. 25,000–20,000 cm$^{-1}$) masked by the CT band and detector saturation in the solid state. The CT band in the solution phase spectrum of **1-Nd** (Supplementary Fig. 14) is slightly more narrow than that of the solid-state, beginning at roughly 480 nm (ca. 20,833 cm$^{-1}$).

Solid-state and solution phase UV–vis–NIR spectra were collected for **1−Nd** (Supplementary Figs. 8 and 15) and **1−Am** (Supplementary Figs. 12 and 18) after 24 h of air exposure. In both states, a substantial decrease in resolution and charge transfer band intensity was observed. Each case presents a notable shift to higher energy as the previously stated splitting degrades. In **1−Nd**, the hypersensitive transition (J = 5/2) at 597 nm (ca. 16,750 cm$^{-1}$) exhibits a hypsochromic shift of roughly 15 nm (ca. 402 cm$^{-1}$). A similar shift of the 527 nm (ca. 18,975 cm$^{-1}$) transition to 509 nm (ca. 19646 cm$^{-1}$) is observed in **1−Am** The 1427 nm (ca. 7007 cm$^{-1}$) transition in **1-Am** is not observed in the solid-state or solution phase. The growth of an intense peak from 450 nm (ca. 22,222 cm$^{-1}$) to 550 nm (ca. 18,182 cm$^{-1}$) was seen in the solution phase of **1-Nd** upon air exposure, but not the

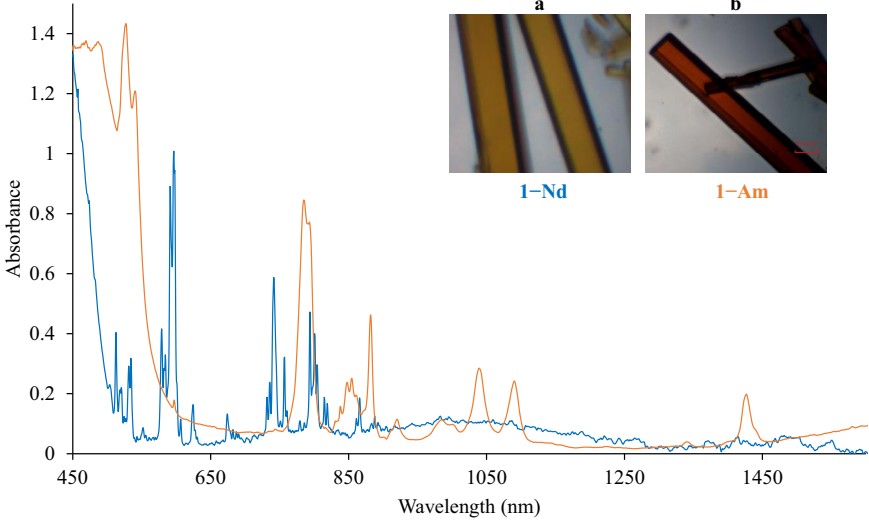

**Fig. 3 Solid-state UV–vis–NIR spectra of 1−Nd (blue) and 1-Am (orange) at 20 °C.** Crystals of **1−Nd** (**a**) and **1−Am** (**b**) used to collect spectra are shown.

<table>
<tr><td colspan="7"><strong>Table 1 QTAIM metrics at the BCP of 1-Nd, 1-Am, and 1-U⋆[39].</strong></td></tr>
<tr><td></td><td colspan="2"><strong>1-Nd</strong></td><td colspan="2"><strong>1-Am</strong></td><td colspan="2"><strong>1-U⋆</strong></td></tr>
<tr><td></td><td><strong>Nd−C$_{avg}$</strong></td><td><strong>Nd−N</strong></td><td><strong>Am−C$_{avg}$</strong></td><td><strong>Am−N</strong></td><td><strong>U−C$_{avg}$</strong></td><td><strong>U−N</strong></td></tr>
<tr><td>$\rho(r)$</td><td>0.2071</td><td>0.2362</td><td>0.2269</td><td>0.2625</td><td>0.2409</td><td>0.2807</td></tr>
<tr><td>$\delta(r)$</td><td>0.1349</td><td>0.1831</td><td>0.1508</td><td>0.2260</td><td>0.1811</td><td>0.2654</td></tr>
<tr><td>$V(r)$</td><td>−461.7</td><td>−554.4</td><td>−582.2</td><td>−759.7</td><td>−574.8</td><td>−754.5</td></tr>
<tr><td>$G(r)$</td><td>462.9</td><td>580.8</td><td>560.1</td><td>761.5</td><td>529.9</td><td>736.9</td></tr>
<tr><td>$H(r)$</td><td>1.2</td><td>26.3</td><td>−22.1</td><td>1.8</td><td>−45.0</td><td>−17.5</td></tr>
<tr><td>$H(r)/\rho(r)$</td><td>7.2</td><td>111.4</td><td>−95.6</td><td>6.7</td><td>−185.1</td><td>−62.5</td></tr>
<tr><td>OS (M)</td><td>3.0</td><td></td><td>2.7</td><td></td><td>3.5</td><td></td></tr>
</table>

The electron density, $\rho(r)$, is given in e Å$^{-3}$; whereas total ($H$) energy density in kJ mol$^{-1}$ Å$^{-3}$. The delocalization index, $\delta(r)$, and integrated oxidation state, OS, are also shown. The latter is obtained as the simple difference between the atomic number of the metal, $Z(M)$, and the localization index, $\lambda(M)$. Fully detailed metrics can be found in Supplementary Tables 6 and 7.

solid-state, likely due to the formation of a precipitate as the compound decomposed.

To get a better understanding of the **1**-**Am** and **1**-**Nd** absorption spectra, the theoretical assignment from the spin–orbit CASSCF/MC-pDFT (SO-pDFT) states was performed in terms of the total angular momentum quantum number, $J$, where the predominant Russell–Saunders terms are indicated accordingly. Given the size of the system, two models were considered, namely Mod1 and Mod2 (see Computational Details). The SO-pDFT states are almost identical for both models with differences no larger than 15 cm$^{-1}$, showing that the fragmentation does not affect the electronic structure significantly.

As shown in Fig. 3 and Supplementary Table 5, transitions in the region comprised between 6863 and 12,738 cm$^{-1}$ (1457–785 nm) that corresponds to $J = 3$–6 (a portion of the $^{7}F_{J}$ manifold, $J = 0$–6) are well predicted. The band starting at ca. 13,158 cm$^{-1}$ (760 nm) with a splitting of ca. 977 cm$^{-1}$ (61 nm) can be ascribed to the characteristic $J = 0$ ($^{7}F_{0}$) $\rightarrow J = 6$ ($^{7}F_{6}$) transition and is predicted with an overestimation of about 814 cm$^{-1}$ (9 nm) and splitting of 977 cm$^{-1}$ (6 nm). Similar procedures have shown errors in the same order magnitude[17,52,53]. Moving to higher energies, the most intense band located between ca. 19,455–18,484 cm$^{-1}$ (514–541 nm) corresponding to the known $J = 0$ ($^{7}F_{0}$) $\rightarrow J = 6$ ($^{5}L_{6}$) transition was predicted to be in the range of 19,729–18,758 cm$^{-1}$ (507–533 nm).

In **1**-**Nd**, the low-lying states corresponding to $J = {}^{9}/_{2}$–$^{15}/_{2}$ ($^{4}I_{J}$ manifold) are not observed in the UV–Vis–NIR spectrum (Fig. 3); however, their prediction agrees with previous theoretical reports[1,54,55]. The peaks located in the region between 575 and 890 nm (ca. 17,391–11,236 cm$^{-1}$) are calculated within an error of ca. 94 cm$^{-1}$ (8 nm) with respect to the experimental observation. Finally, the energy of the most intense band corresponding to the hypersensitive $^{4}I_{9/2} \rightarrow {}^{4}G_{5/2}$ transition is in the range of 17,421–16,367 cm$^{-1}$ (574–611 nm) and was predicted between ca. 17,547–16,714 cm$^{-1}$ (570–598 nm) (Supplementary Table 6).

Despite the good agreement between the experimental and theoretical results, it is important to highlight the role of the dynamic correlation in the correct determination of these energies. As already reported in other americium(III) and neodymium(III) systems, at higher energies an overestimation of the $f$–$f$ transitions are observed[52,53]. The importance of dynamical correlation in our systems is reflected in the most intense transitions. If we consider the band observed at 514 nm in **1**-**Am**, the difference between SO-CAS (395 nm) and SO-pDFT (516 nm) is 5820 cm$^{-1}$ (Supplementary Tables 5 and 6) showing the increased accuracy achieved by considering dynamical correlation.

In order to better understand the nature of the chemical bond in **1**-**Nd** and **1**-**Am**, we applied Bader's quantum theory of atoms in molecules (QTAIM)[56]. Within this formalism, the chemical bond is analyzed through metrics such as accumulation of electron density, $\rho(r)$, energy densities (potential $V(r)$, kinetic $G(r)$, and total $H(r)$), and localization, $\lambda(M)$, and delocalization $\delta(r)$, indices at the so-called bond critical point (BCP) (see Computational details). These BCPs are the regions along the bond paths where the electron density reaches a minimum (saddle point) and the interaction forces cancel out. Table 1 summarizes the main QTAIM metrics including the M−Cp' bonds as an average interaction. Individual M−C bond metrics are found in the Supplementary Information (Supplementary Tables 7–9).

The interaction of $f$-block elements and cyclopentadienyl ligands has been vastly investigated but it is still subject to debate. It is clear that all five carbon atoms in the Cp′ unit are involved in rather weak interactions with low degrees of covalency[57,58]. Deviations from an equidistant interaction such as in the M−Cp bonds are observed when a fourth ligand coordinates and/or a Cp derivative is used. The classic arrangement for the coordination of

three Cp′ ligands to *f*-block ions corresponds to two –SiMe₃ groups facing up and the third facing down[20]. Therefore, the combination of coordinating Cp' ligands and a fourth bulky ligand such as 4,4′-bpy makes the number of BCPs <5 (Supplementary Fig. 28), implying at a first glance that the interaction is weaker than in more symmetrical M−Cp interactions. If we now consider the QTAIM metrics for the BCPs found, it is possible to note that in both **1-Nd** and **1-Am** the M−N interactions are stronger than the M−C bonds, as noted in the increased $\rho(r)$ and $\delta(r)$ values. However, it is noteworthy that this does not imply greater covalency in those bonds. To be able to estimate covalent contributions, the total energy density must have a negative sign because of the stabilization of the concentration of electron density at the BCP by an excess of potential energy density $V(r)$. The opposite is true for ionic bonds where an excess of kinetic energy density destabilizes $\rho(r)$ at the BCP, indicating that electrostatic interactions are responsible for the bond formation. The results indicate that Nd³⁺ in **1-Nd** binds ionically to both 4, 4′−bpy and Cp′ ligands with an integrated oxidation state of 3.00 (Table 1 and Supplementary Table 7). The excess of $G(r)$ is more pronounced in Nd−N bonds, which is unusual for the nature of 4,4′-bpy. Conversely and as expected, Am−C bonds display a certain degree of covalency $(H(r)/\rho(r) = -95.6\ \text{kJ mol}^{-1}\ \text{Å}^{-3})$ which agrees with previously reported values for AmCp₃ $(H(r)/\rho(r) = -81.6\ \text{kJ mol}^{-1}\ \text{Å}^{-3})$[57]. Surprisingly, the Am−N bond is ionic despite pyridine-based ligands' low energy $\pi^*$ orbitals ability to mix with the occupied Am³⁺ 5*f* electrons to form π-backbonds[1,59]. It is very likely that the origin resides in the steric effects imposed by the −SiMe₃ groups blocking the "full" coordination of 4,4′-bpy, which is also supported by the longer bond lengths observed for Am−N bonds and Am−Cent discussed in the structural analysis. Another interesting feature in **1−Am** is that the integrated oxidation state is lower (2.7+) than the formal 3+ value, consistent with the forward donation of Cp' ligands to the metal and absence of π-back donation (or back bonding). These metrics can be contrasted with the previously reported uranium analog, **1-U\***[39]. U−Cent (average) bonds are clearly more covalent than those of **1-Am** based on the $H(r)$ values (185.1 kJ mol⁻¹ Å⁻³) and more BCPs were found compared to both **1-Nd** and **1-Am** indicating a more defined topology of the electron density in the U−Cp' interatomic region. Unlike **1-Nd** and **1-Am**, the U−N bond in **1-U\*** displays a negative $H(r)$ revealing the increased covalent character in all the U–ligand bonds. This may be attributed to the fact that uranium is a larger ion compared to americium, thus reducing the impact of steric hindrance of the ligands and increasing the ability of the metal to interact more effectively with the N and Cent atoms. Another interesting difference is that the integrated oxidation state is significantly increased for U (3.5+), but this might be due to the difference in their corresponding (II/III) and (III/IV) redox potentials rather than the nature of the chemical bond. Thus, while Nd(III) and Am(III) are more prone to accept electron density from the ligands, U(III) donates it to the local environment increasing the integrated oxidation state.

In summary, multinuclear organometallic americium, (Cp′₃Am)₂ (*µ*-4,4′-bpy), **1-Am**, and its neodymium analogue, (Cp′₃Nd)₂(*µ*-4, 4′-bpy), **1-Nd**, were characterized by single-crystal X-ray diffraction, in addition to solution and solid-phase UV–vis–NIR absorption spectroscopy. The average M−Cent in **1-Am** is slightly longer than that observed in the literature as a direct result of steric hindrance between the coordinated 4,4′-bpy and Cp′ ligands. The steric factor becomes less important in the uranium analog due to its increased ionic radius, which displays significantly larger covalent contributions to the chemical bond. The competitive coordination environment between 4,4′-bpy and Cp′ leads to a surprisingly ionic Am−N bond,

greater in length than previously anticipated. Nd−C bonds in **1-Nd** were determined to be ionic in nature, deviating from the expected covalent character generally observed in Ln−Cp complexes. Coordination of Cp′ and 4,4′-bpy gives rise to evident splitting and a bathochromic shift of fingerprint *f–f* transitions consistent with those commonly observed in organometallic systems.

## Methods

Caution! ²⁴³Am $(t_{1/2} = 7364\ \text{years})$ possesses significant health and radiological hazards resulting from α- and γ-emission, and also due to a short-lived, high energy β- and γ-emitting ²³⁹Np $(t_{1/2} = 2.356\ \text{days})$ daughter product. All handling of ²⁴³Am was completed with proper controls in a HEPA filter-equipped Category II radiologic facility.

C₆D₆ (Cambridge), CD₂Cl₂ (Cambridge), and THF-*d₈* (Cambridge) were freeze-pump-thawed (3×) over activated neutral alumina. C₆D₆ and THF-*d₈* were stored over NaK and CD₂Cl₂ were stored over activated 3 Å molecular sieves prior to use. Diethyl ether (Sigma), dimethoxyethane (Sigma), and hexane (Sigma) were distilled from sodium benzophenone ketyl (Sigma) and stored over activated 3 Å molecular sieves (Sigma). Toluene (Sigma) was sparged with argon, run through columns of molecular sieves and Q-5, and stored over activated 3 Å molecular sieves. Additionally, diethyl ether, hexane, and toluene were stored over NaK 48 h prior to use. Dimethoxyethane (Sigma) was stored over activated neutral alumina (Sigma) for 48 h prior to use. Dichloromethane (Sigma) for Evans Method measurements was used as received. Distilled H₂O, hydrobromic acid (8.77 M, Sigma), Hydrofluoric Acid (28.9 M, Sigma), potassium *bis*(trimethylsilyl)amide (95%, Sigma), bromo-trimethylsilane (97%, Sigma), chlorotrimethylsilane (98%, Sigma), anhydrous NdCl₃ (Sigma) and 4,4′−bpy (98%, Sigma) were used as received.

All reactions were performed with rigorous exclusion of air and water under an argon atmosphere utilizing Schlenk line and glovebox techniques, except where noted. Reactions with ²⁴³Am were performed in a dedicated, well-ventilated HEPA filtered fume hood or glovebox.

**KCp′**. KCp (3.858 g, 0.037 mol) was stirred with chlorotrimethylsilane (4.844 mL, 0.037 mol) for 4 h in diethyl ether (200 mL), followed by filtering off the resulting KCl byproduct on a frit. Potassium *bis*(trimethylsilyl)amide (7.016 g, 0.035 mol) was added to the resulting solution and stirred overnight. Diethyl ether was removed under reduced pressure and hexane (200 mL) was added to the resulting oil. The slurry was stirred and filtered, isolating KCp′. The fine powder was washed with hexane (3 × 20 mL) and dried under reduced pressure overnight[60,61].

**Cp′₃Nd**. NdCl₃ (50 mg, 0.200 mmol) and a slight excess of KCp′ (122.3 mg, 0.693 mmol) were stirred vigorously in diethyl ether (2 mL) overnight. The resulting green slurry was centrifuged and filtered, followed by washing the pellet with diethyl ether (3 × 1 mL). Diethyl ether was removed under reduced pressure and hexane (2 mL) was added in order to remove unreacted KCp′. The slurry was again centrifuged and filtered, followed by washing the pellet with hexane (3 × 1 mL). The resulting green powder was dried under reduced pressure overnight[36].

**(Cp′₃Nd)₂(µ−4,4′−bpy), 1−Nd**. Mint green Cp′₃Nd (20 mg, 0.035 mmol) and colorless 4,4′-bpy (3 mg, 0.02 mmol) were combined in toluene (2 mL) and stirred overnight yielding a lime green slurry. Toluene was evaporated under reduced pressure and the resulting powder was rinsed with hexane to remove unreacted Cp′₃Nd then dried under reduced pressure. The powder was again slurried in toluene, heated to a gentle boil, and filtered. Slow cooling to room temperature over 1 h, followed by standing at room temperature for 1 h yielded green crystals suitable for single-crystal X-ray diffraction (20.7 mg, 0.016 mmol, 91% yield)[1].H NMR (600 MHz, CD₂Cl₂): δ −4.93 (*s*, Cp′−H, 12H), 3.03 (*s*, TMS, 54H), 9.03 (*dd*, bpy −H, 4H), 9.11 (*dd*, bpy−H, 4H), 13.61 (*s*, Cp′−H, 12H) (Supplementary Fig. 20). Magnetic susceptibility (Evans Method): $\mu_{\text{eff}} = 3.47\mu_B$ (Supplementary Fig. 24). UV–vis–NIR (toluene): $\lambda_{\text{max}}$ nm (cm⁻¹) = 513 (19,493), 520 (19,231), 531 (18,832), 534 (18,727), 539 (18,553), 542 (18,450), 553 (18,083), 579 (17,271), 584 (17,123), 592 (16,892), 597 (16,750), 600 (16,667), 606 (16,502), 613 (16,313), 618 (16,181), 625 (16,000), 674 (14,837), 682 (14,663), 687 (14,556), 690 (14,493), 701 (14,265), 732 (13,661), 735 (13,605), 742 (13,477), 751 (13,316), 757 (13,210), 761 (13,141), 770 (12,987), 776 (12,887), 780 (12,821), 786 (12,723), 794 (12,594), 801 (12,484), 804 (12,438), 815 (12,270), 819 (12,210), 825 (12,121), 829 (12,063), 835 (11,976), 861 (11,614), 867 (11,534), 883 (11,325), 889 (11,249), 893 (11,198), 911 (10,977) (Supplementary Fig. 6).

**(Cp′₃Am)₂(µ-4,4′-bpy), 1-Am**. In a fume hood, an aliquot of Am³⁺ (5 mg, 0.02 mmol metal content) was drawn from a stock solution in HCl (2 M), dried to a powder, and taken up in water (1 mL). Excess NH₄OH (~3 mL, 14.8 M) was added to the solution. The resulting Am(OH)₃·nH₂O precipitate was centrifuged and washed with water (3 × 2 mL) to remove excess NH₄Cl. The pellet was dissolved in HBr (2 mL, 8.77 M) and HF (5 µL, 28.9 M), then transferred to a 20 mL scintillation

vial where it was evaporated to a residue under a heat lamp and stream of house $N_{2(g)}$. The resulting $AmBr_3 \cdot nH_2O$ residue was washed with diethyl ether until the washings were colorless. The peach-colored powder was pumped into the glovebox overnight.

Inside the glove box, in a manner similar to that of $AmCl_3(DME)_n$[17], dimethoxyethane (DME, 2 mL) was added to the $AmBr_3 \cdot nH_2O$ and stirred for 10 min. Bromotrimethylsilane (TMS—Br, 2 mL) was added dropwise to the solution and stirred at 50 °C for 2 h. The slurry was cooled to room temperature and hexane (3 mL) was added. After settling, the colorless supernatant was carefully pipetted away, the tan powder was washed with hexane (3 × 3 mL), and the sample was dried under reduced pressure for 30 min. Ether (2 mL) was added and the slurry was stirred for 15 min, followed by the addition of hexane (3 mL). The colorless supernatant was removed after allowing the slurry to settle and the product was dried under reduced pressure for 3 h.

The resulting $AmBr_3(DME)_n$ was slurried in diethyl ether (1 mL). KCp′ (12 mg, 0.07 mmol) in diethyl ether (1 mL) was added dropwise to the solution and stirred vigorously for 30 min. A gradual color change from peach to rose gold was observed, as well as the generation of a colorless precipitate, presumably KBr.

The solids were removed by centrifugation and washed with ether (3 × 1 mL), and the collected supernatants were filtered and dried under reduced pressure. The product was dissolved in hexane (2 mL) and centrifuged to remove the white precipitate, presumably excess KCp′, and washed with hexane (3 × 1 mL). The solution was filtered and dried under reduced pressure, yielding a pale, rose gold-colored solid, presumably Cp′$_3$Am.

The resulting Cp′$_3$Am was dissolved a minimal amount of toluene. 4,4′-bpy (1.6 mg, 0.01 mmol) was dissolved in toluene (1 mL), added dropwise to the putative Cp′$_3$Am solution, and stirred vigorously resulting in a bright orange solution. The solution was concentrated under reduced pressure until an orange precipitate, (Cp′$_3$Am)$_2$($\mu$-4,4′-bpy), **1-Am**, was observed. The slurry was heated to 120 °C while gently stirring until all material was dissolved. The solution was slowly cooled to room temperature without stirring over 1 h. Bright orange block crystals suitable for single-crystal X-ray diffraction were grown at room temperature overnight[1].H NMR (600 MHz, $C_6D_6$): δ 1.34 (s, TMS, 54H), 7.02 (dd, bpy—H, 4H), 7.13 (dd, bpy—H, 4H), 8.53 (s, Cp—H, 12H), 12.44 (s, Cp—H, 4H) (Supplementary Fig. 22). UV–vis–NIR (toluene): $\lambda_{max}$ nm (cm$^{-1}$) = 527 (18,975), 540 (18,519), 597 (16,750), 785 (12,739), 794 (12,594), 831 (12,034), 838 (11,933), 848 (11,792), 855 (11,696), 861 (11,614), 882 (11,338), 920 (10,870), 985 (10,152), 1002 (9980), 1038 (9634), 1090 (9174), 1340 (7463), 1427 (7008) (Supplementary Fig. 10).

## Data availability

Further data for this study is available in the Supplementary Information. Structural data for **1—Nd** and **1—Am** is available in the Cambridge Structural Database under deposition numbers 2081956 and 2081957, respectively. Source data are provided with this paper.

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

## Acknowledgements

We would like to thank the Department of Energy Office of Science, Basic Energy Sciences, DOE-BES Heavy Elements Chemistry for funding under award DE-FG02-13ER16414 (T.E.A.-S.). A special thank you goes to our radiation safety officers, Ashley Gray and Jason Johnson, as well as Oak Ridge National Laboratory for supplying the isotopes used in this study. We would also like to extend our gratitude to the William J. Evans group at the University of California Irvine for supplying Cp′ used in reaction optimization.

## Author contributions

B.N.L. completed synthetic work, crystallography, solution-phase absorption spectroscopy, solution-phase magnetic susceptibility, and NMR. M.J.B.L. and C.C.B. completed all theoretical work including CASSCF and QTAIM analysis. J.M.S. and T.N.P. completed solid-state absorption spectroscopy as well as assisted with synthetic work and actinide purification. R.E.B. completed solid-state magnetism experiments. C.J.W. assisted with starting material syntheses and proposed 4,4′-bpy as bridging ligand. T.E.A.-S. proposed multinuclear americium and was project overseer. All authors contributed and have approved the submission of the manuscript.

## Competing interests

The authors declare no competing interests.
