## [Peer Review File · Nature Communications]

Cyclopentadienyl Coordination Induces Unexpected Ionic Am –N Bonding in an Americium Bipyridyl ComplexREVIEWER COMMENTS

Reviewer #1 (Remarks to the Author):

The paper by Long et al. describes the reactivity of two tris-cyclopentadienyl complexes, $\text{Nd}(\text{Cp}')_3$ and $\text{Am}(\text{Cp}')_3$ ($\text{Cp}' = \{\text{C}_5\text{H}_4\text{SiMe}_3\}^-$) with 4,4'-bipyridine, resulting in the formation of bimetallic dimers of formula $[\{\text{M}(\text{Cp}')_3\}_2(\mu\text{-}4,4'\text{-bipy})]$ (1-M; M = Nd, Am). Organometallic chemistry of Am is hugely underdeveloped, with the first example of a structurally authentic Am organometallic reported in 2019 by Goodwin et al (ACIE 2019, 58, 11695). Crucially, 1-Am is only the second example of a structurally authenticated Am organometallic complex and the very first example of a bimetallic system. The experimental work is very solid and rigorous. The electronic structure of the compounds has been very thoroughly characterized and the experimental results are strongly complemented by computational analysis. The authors probed subtle effects on the electronic structure provoked by the coordination of the bipy ligand, leading to some unexpected results in terms of the extent of ionic contribution to the Am-Nbipy bonding interactions. However, what is striking is the absence of magnetic measurements (solution Evans method, SQUID and EPR); this is a bit of a shame, as simple magnetic measurements could have been performed at least on 1-Nd. These measurements would have significantly elevated the paper, but it doesn't take away from the fact that this is a very good piece of work indeed.

In summary, this is an excellent piece of synthetic work, which is further enriched by the use of computational techniques and will be of great significance for the f-element community and, more in general, those with a keen interest in actinide and transuranic chemistry – especially considering that there aren't that many Am organometallic compounds around, and I suspect 1-Am is potentially the first of a whole new family of multimetallic transuranic compounds. Therefore, I believe this work would be perfectly suited for the readership of Nature Communications. There are however some points which I would kindly ask the authors to consider and address prior to publication.

1) The authors should do some comparisons of M-N and M-Cp distances in 1-Am and 1-Nd with the following complexes:

- $[\{\text{UCp}'_3\}_2(\mu\text{-}4,4'\text{-bipy})]$ and $[\{\text{ThCp}'_3\}_2(\mu\text{-}4,4'\text{-bipy})]$ (both already referenced in the manuscript); comparison of metrics would make some of the observations regarding the Am-N and Am-Cp interactions more robust – the Th derivative is a tetravalent species with di-reduced bipy, but comparisons are still relevant nonetheless.

- $[\{\text{CeCp}'_3\}_2(\mu\text{-}4,4'\text{-bipy})]$ (Dalton Trans 2004 579).

2) The ^1H NMR spectrum of 1-Am looks very clean and signals are relatively well resolved; I believe it would be possible to attempt a magnetic susceptibility measurement via Evans method with a $\text{C}_6\text{H}_6/\text{C}_6\text{D}_6$ insert. If doable, this would complement well the characterization of the ground state of 1-Am (but I do fully appreciate this might not be possible to do because of resources and technical complications). This should at least be attempted for 1-Nd and its precursor $\text{Nd}(\text{Cp}')_3$ to probe the influence of the bipy ligand on the ground state – which should be a good starting point for future work on related actinide and transuranic congeners.

3) I believe the paper would be elevated further if the authors performed QAIM analysis of Ephritikhine's $[\{\text{UCp}'_3\}_2(\mu\text{-}4,4'\text{-bipy})]$. The authors rightly point out that $\text{Ln}(\text{Cp})_3$ and $\text{An}(\text{Cp})_3$ bonding has been very thoroughly investigated, but they are now in possession of a unique structural motif for Am and additional comparisons with the U analogue would be extremely insightful. QAIM analysis was already performed on Mills' $[\{\text{ThCp}'_3\}_2(\mu\text{-}4,4'\text{-bipy})]$, so that could also be added to the discussion.

4) It's interesting that the integrated OS of 1-Am is slightly lower than that of 1-Nd. Magnetic susceptibility measurements (see above) might provide some insights on this. Have the authors attempted to measure electrochemical properties or chemically reduce 1-Am and 1-Nd? Though it's likely that the π^* manifold of the bipy ligand will be reduced first, it would be interesting to see differences in the electronic spectra as this could possibly result in a stronger interaction with the

metal centre, which in turn will also affect M-Cp bonding. I don't know if this is something achievable when working with Am (I appreciate there could be problems with contamination of electrochemical equipment) and these systems in general, so I would kindly ask the authors to comment on this. Additionally, is this something that could be, in principle, tentatively modelled computationally a priori?

5) The structural characterization is thorough, however it gets a bit confusing without a good reference, and unfortunately the two crystallography tables included in the SI (Tables 2 and 4) aren't particularly good. The authors should improve this and compile together relevant metrics of 1-Am and 1-Nd in the same table to facilitate analysis (and help the readers). Addition of Ephritikhine's U and Ce analogues (see my previous point) to this table would also be beneficial.

6) 'Synthesis' section – the authors should report here NMR peaks and UV-vis-NIR absorptions for both 1-Nd and 1-Am.

Reviewer #2 (Remarks to the Author):

The manuscript under consideration describes the synthesis of an Am organometallic complex along with its Nd homologue. Both complexes are characterized spectroscopically, structurally using single crystal X-ray diffraction with complementary computational studies of the optical spectra and bonding using QTAIM analysis.

The experimental work is complete and robust. However, the manuscript should be revised to more emphatically or convincingly support the authors conclusions and statements regarding the novelty of the reported complex and its importance to advancing f-element chemistry.

The novelty of the paper is the synthesis of the complex, but the results of the computational and spectroscopic studies do not add any new insight into actinide bonding. In fact, the references cited highlight a low-degree of covalent bonding in these types (An-Cp) of complexes (ref 44 and 45 as cited on line 207) and the authors' current study only reiterates earlier studies of actinide cyclopentadienyl complexes.

Overall, the introduction of the paper has little relevance to the issues of f-element bonding in a direct way. It is mostly concerned with Cp' chemistry and the lack of transuranium complexes reported with Cp'. The entire effort of the authors to highlight the importance of the reported complex would be strengthened by refocusing the introduction to align with the conclusions of the paper as well as the title.

There is nothing technically problematic with the paper. However, this work would be better served by a revision of the manuscript highlighting its significance to f-element chemistry, rather than the herculean efforts of its synthesis and preparation.

Reviewer #3 (Remarks to the Author):

This paper describes only the 2nd example ever of an organometallic Am compound and the first Am bimetallic. To work with Am organometallics is exceptionally difficult and the authors are to be congratulated on the experimental data collected for Am. This work of great significance to the field. I do recommend publication as a Nature Communication, but only after major revision of the manuscript.

This is because while the experimental data for Am are convincing (the standard characterisation data for the Nd analogue are less so), the level of data analysis is not. I do not dispute that the spectroscopy of the Am complex is 'unique' and of course the first Am multimetallic is an important

opportunity to learn about the nature of organometallic Am bonding. Theory is likewise needed for assignment of electronic structure and to provide insight into the nature of bonding.

I am not convinced by the theoretical work as it stands. I also do not think that it has been adequately demonstrated that there is good agreement between experiment and theory. It reads to me that the changes observed experimentally (SCXRD and UV-vis-NIR) have been overinterpreted to fit with the theory, to enable the authors to claim significant changes to the ionic character of bonding or oxidation state, which I remain unconvinced by.

November 8, 2021

Response to reviewers

We would like to thank the reviewers for their suggestions, each of which has greatly improved the quality of the manuscript. Below are the specific responses to reviewer comments and how they were addressed in the manuscript:

Reviewer #1 (Remarks to the Author):

The paper by Long et al. describes the reactivity of two tris-cyclopentadienyl complexes, $\text{Nd}(\text{Cp}')_3$ and $\text{Am}(\text{Cp}')_3$ ($\text{Cp}' = \{\text{C}_5\text{H}_4\text{SiMe}_3\}^-$) with 4,4-bipyridine, resulting in the formation of bimetallic dimers of formula $[\{\text{M}(\text{Cp}')_3\}_2(\mu\text{-4,4'-bipy})]$ (1-M; M = Nd, Am). Organometallic chemistry of Am is hugely underdeveloped, with the first example of a structurally authentic Am organometallic reported in 2019 by Goodwin et al (ACIE 2019, 58, 11695). Crucially, 1-Am is only the second example of a structurally authenticated Am organometallic complex and the very first example of a bimetallic system. The experimental work is very solid and rigorous. The electronic structure of the compounds has been very thoroughly characterized and the experimental results are strongly complemented by computational analysis. The authors probed subtle effects on the electronic structure provoked by the coordination of the bipy ligand, leading to some unexpected results in terms of the extent of ionic contribution to the Am-Nbipy bonding interactions. However, what is striking is the absence of magnetic measurements (solution Evans method, SQUID and EPR); this is a bit of a shame, as simple magnetic measurements could have been performed at least on 1-Nd. These measurements would have significantly elevated the paper, but it doesn't take away from the fact that this is a very good piece of work indeed.

In summary, this is an excellent piece of synthetic work, which is further enriched by the use of computational techniques and will be of great significance for the f-element community and, more in general, those with a keen interest in actinide and transuranic chemistry – especially considering that there aren't that many Am organometallic compounds around, and I suspect 1-Am is potentially the first of a whole new family of multimetallic transuranic compounds. Therefore, I believe this work would be perfectly suited for the readership of Nature Communications. There

are however some points which I would kindly ask the authors to consider and address prior to publication.

1) The authors should do some comparisons of M-N and M-Cp distances in 1-Am and 1-Nd with the following complexes:

- $[(UCp'_{3})_{2}(\mu-4,4'-bipy)]$ and $[(ThCp'_{3})_{2}(\mu-4,4'-bipy)]$ (both already referenced in the manuscript); comparison of metrics would make some of the observations regarding the Am-N and Am-Cp interactions more robust – the Th derivative is a tetravalent species with di-reduced bipy, but comparisons are still relevant nonetheless.

- $[(CeCp'_{3})_{2}(\mu-4,4'-bipy)]$ (Dalton Trans 2004 579).

***Response:** Structural comparison to the suggested complexes has been added to the manuscript, as well as additional in-depth comparison in **Table S4** of the supplementary information.*

2) The 1H NMR spectrum of 1-Am looks very clean and signals are relatively well resolved; I believe it would be possible to attempt a magnetic susceptibility measurement via Evans method with a C_6H_6/C_6D_6 insert. If doable, this would complement well the characterization of the ground state of 1-Am (but I do fully appreciate this might not be possible to do because of resources and technical complications). This should at least be attempted for 1-Nd and its precursor $Nd(Cp')_3$ to probe the influence of the bipy ligand on the ground state – which should be a good starting point for future work on related actinide and transuranic congeners.

***Response:** Magnetic susceptibility measurements were completed on $NdCp'_3$ and $(Cp'_3Nd)_2(\mu-4,4'-bpy)$ via the Evans Method. Additionally, magnetic Susceptibility measurements were completed on crystals of 1-Nd and added to the manuscript, as well as **Figure S23** of the supplementary information.*

Americium(III) we have measured in the past have generally been strongly nonmagnetic. The difficulties in loading an air-sensitive solid-state sample to transfer to the National High Magnetic Field Laboratory where our samples are measured for a sample expected to be strongly nonmagnetic outweigh the benefits. Similarly, solubility issues in available glovebox solvents and low yield after removing crystal samples results in too little material suitable for the Evan's

Method. So, we appreciate the reviewer's comment but unfortunately have to decline his suggestion.

3) I believe the paper would be elevated further if the authors performed QTAIM analysis of Ephritikhine's $[(UCp'_3)_2(\mu-4,4'-bipy)]$. The authors rightly point out that $Ln(Cp)_3$ and $An(Cp)_3$ bonding has been very thoroughly investigated, but they are now in possession of a unique structural motif for Am and additional comparisons with the U analogue would be extremely insightful. QTAIM analysis was already performed on Mills' $[(ThCp''_3)_2(\mu-4,4'-bipy)]$, so that could also be added to the discussion.

*Response: We thank the reviewer for the suggestion. We have calculated $(UCp'_3)_2(\mu-4,4'-bipy)$ from the experimental crystal structure and discussed the results in the main text. Unfortunately, including Th to the comparison would be meaningless due to the QTAIM values reported in the article for the Th analog are likely to be overestimated due to the lack of static correlation in the molecular electron density. However, adding U to the discussion helped to highlight the unexpected low degree of covalency in **1-Am**.*

4) It's interesting that the integrated OS of 1-Am is slightly lower than that of 1-Nd. Magnetic susceptibility measurements (see above) might provide some insights on this. Have the authors attempted to measure electrochemical properties or chemically reduce 1-Am and 1-Nd? Though it's likely that the π^* manifold of the bipy ligand will be reduced first, it would be interesting to see differences in the electronic spectra as this could possibly result in a stronger interaction with the metal centre, which in turn will also affect M-Cp bonding. I don't know if this is something achievable when working with Am (I appreciate there could be problems with contamination of electrochemical equipment) and these systems in general, so I would kindly ask the authors to comment on this. Additionally, is this something that could be, in principle, tentatively modelled computationally a priori?

Response: This is an interesting subject indeed. We have not attempted any electrochemical nor chemical reduction on 1-Am so far, but it is a short term goal. We have performed computations to predict the electronic structure of the reduced species, and despite the variations on the predicted ground state (high or low spin) with the method used, the spin density (see figures) suggests that both, the bipy and metal are partially reduced. This true for Am and Nd from the computational predictions, but we have been able to chemically reduce 1-Nd and 1-Sm and the

UV-vis spectra displays peaks corresponding to both Ln(II) and Ln(III) indicating an itinerant electron in line with the computational results. We share these preliminary data in good faith with the reviewer to show that this is a paper in progress that we expect to submit within a year.

5) The structural characterization is thorough, however it gets a bit confusing without a good reference, and unfortunately the two crystallography tables included in the SI (Tables 2 and 4) aren't particularly good. The authors should improve this and compile together relevant metrics of 1-Am and 1-Nd in the same table to facilitate analysis (and help the readers). Addition of Ephritikhine's U and Ce analogues (see my previous point) to this table would also be beneficial.

Response: *The crystallographic tables have been reworked for easier comparison, including the addition of similar published systems with Ce, Th, and U complexes mentioned in comment 1.*

6) 'Synthesis' section – the authors should report here NMR peaks and UV-vis-NIR absorptions for both 1-Nd and 1-Am.

Response: NMR peaks and UV-vis-NIR absorption transitions for 1-Nd and 1-Am have been added to the synthesis section.

Reviewer #2 (Remarks to the Author):

The manuscript under consideration describes the synthesis of an Am organometallic complex along with its Nd homologue. Both complexes are characterized spectroscopically, structurally using single crystal X-ray diffraction with complementary computational studies of the optical spectra and bonding using QTAIM analysis. The experimental work is complete and robust. However, the manuscript should be revised to more emphatically or convincingly support the authors conclusions and statements regarding the novelty of the reported complex and its importance to advancing f-element chemistry.

The novelty of the paper is the synthesis of the complex, but the results of the computational and spectroscopic studies do not add any new insight into actinide bonding. In fact, the references cited highlight a low-degree of covalent bonding in these types (An-Cp) of complexes (ref 44 and 45 as cited on line 207) and the authors' current study only reiterates earlier studies of actinide cyclopentadienyl complexes.

Overall, the introduction of the paper has little relevance to the issues of f-element bonding in a direct way. It is mostly concerned with Cp' chemistry and the lack of transuranium complexes reported with Cp'. The entire effort of the authors to highlight the importance of the reported complex would be strengthened by refocusing the introduction to align with the conclusions of the paper as well as the title.

There is nothing technically problematic with the paper. However, this work would be better served by a revision of the manuscript highlighting its significance to f-element chemistry, rather than the herculean efforts of its synthesis and preparation.

Response: We thank the reviewer for the suggestion. The introduction of the manuscript has been revised to focus on the importance of bonding and how our contribution provides further insight into this matter.

Reviewer #3 (Remarks to the Author):

This paper describes only the 2nd example ever of an organometallic Am compound and the first Am bimetallic. To work with Am organometallics is exceptionally difficult and the authors are to be congratulated on the experimental data collected for Am. This work of great significance to the field. I do recommend publication as a Nature Communication, but only after major revision of the manuscript.

This is because while the experimental data for Am are convincing (the standard characterisation data for the Nd analogue are less so), the level of data analysis is not. I do not dispute that the spectroscopy of the Am complex is 'unique' and of course the first Am multimetallic is an important opportunity to learn about the nature of organometallic Am bonding. Theory is likewise needed for assignment of electronic structure and to provide insight into the nature of bonding.

I am not convinced by the theoretical work as it stands. I also do not think that it has been adequately demonstrated that there is good agreement between experiment and theory. It reads to me that the changes observed experimentally (SCXRD and UV-vis-NIR) have been overinterpreted to fit with the theory, to enable the authors to claim significant changes to the ionic character of bonding or oxidation state, which I remain unconvinced by.

Response: *We agree with the reviewer that the theoretical work was not as precise as it should be. The reason is that the first time we did not include dynamic correlation to correct the energy of the spin – orbit states, which led to non – negligible errors in the high – energy region of the spectra. However, we have fixed this problem by running multiconfigurational pair – density functional theory (MC-pDFT) which accounts for the dynamic correlation. Therefore, the error in the predictions were significantly reduced thus improving the agreement between theory and experiment. Since the disagreement mostly lied in the high-energy region, the nature of the chemical bond should not be affected. Therefore, the calculated ground-state molecular electron density does not need improvement and the metrics derived from it are still valid.*

REVIEWER COMMENTS

Reviewer #1 (Remarks to the Author):

I would like to thank the authors for addressing all the points raised in the original review. The additional experimental and theoretical work are more than satisfactory and address also other reviewers' comments. Therefore, I believe the paper should be accepted.

I have just a couple of small comments:

- In my original review I made a mistake when referring to the Ephritikhine's Ce-bipy adduct (the compound is in fact a pyridine adduct and the authors were very good at rectifying this). Since then, I have also discovered a PhD thesis from University of California Berkley, which happens to be relevant to this work in some respect. The author is Daniel Kazhdan from Richard Andersen's group and they report the structure of the dimer $(\text{CpMe})_3\text{Ce}(4,4'\text{-bipy})\text{Ce}(\text{CpMe})_3$. I appreciate this is not peer reviewed work, but the authors might want to consider adding this (maybe to the SI XRD table). This wasn't originally mentioned in my first review, so I am not expecting the authors to address this, it's more of a suggestion.

- In the new crystallography table (Table S4), there is a typo in the first entry of the Nd column: 2668.(5) - which I believe should be 2.668(5)

Reviewer #3 (Remarks to the Author):

I agree with the authors that the review process has improved the paper and accept their responses to my review. I am therefore happy to recommend publication.

December 9, 2021

Response to reviewers

We would like to thank the reviewer for their suggestion and have implemented it to manuscript accordingly.

Reviewer #1 (Remarks to the Author):

I would like to thank the authors for addressing all the points raised in the original review. The additional experimental and theoretical work are more than satisfactory and address also other reviewers' comments. Therefore, I believe the paper should be accepted.

I have just a couple of small comments:

- In my original review I made a mistake when referring to the Ephritikhine's Ce-bipy adduct (the compound is in fact a pyridine adduct and the authors were very good at rectifying this). Since then, I have also discovered a PhD thesis from University of California Berkley, which happens to be relevant to this work in some respect. The author is Daniel Kazhdan from Richard Andersen's group and they report the structure of the dimer $(\text{CpMe})_3\text{Ce}(4,4'\text{-bipy})\text{Ce}(\text{CpMe})_3$. I appreciate this is not peer reviewed work, but the authors might want to consider adding this (maybe to the SI XRD table). This wasn't originally mentioned in my first review, so I am not expecting the authors to address this, it's more of a suggestion.

Response: Data from the suggested structure has been added to the bond length table in the supplementary information and referenced accordingly.

- In the new crystallography table (Table S4), there is a typo in the first entry of the Nd column: 2668.(5) - which I believe should be 2.668(5)

Response: The typing error has been fixed.